# Spanish version of Multidimensional Mentalizing Questionnaire (MMQ): Translation, adaptation and psychometric properties in a Chilean population

**Nerea Aldunate**[1], **Pablo López-Silva**[2,3], **Cristian Brotfeld**[4], **Ernesto Guerra**[5], **Edmundo Kronmüller**[6]*

**1** Laboratorio de Comportamiento Animal y Humano, Centro de Investigación en Complejidad Social, Facultad de Gobierno, Universidad del Desarrollo, Santiago, Chile, **2** Escuela de Psicología, Universidad de Valparaíso, Valparaíso, Chile, **3** Instituto de Sistemas Complejos de Valparaíso, Valparaíso, Chile, **4** Fundación Astoreca, Santiago, Chile, **5** Centro de Investigación Avanzada en Educación, Instituto de Educación, Universidad de Chile, Santiago, Chile, **6** Escuela de Psicología, Pontificia Universidad Católica de Chile, Santiago, Chile

* ekr@uc.cl

**Data Availability Statement:** Open Data and Scripts: The information needed to reproduce all of

## Abstract

This paper presents the first translation and adaptation of the Multidimensional Mentalizing Questionnaire (MMQ) into Spanish for a native Spanish-speaking sample in Chile. The study examines the psychometric properties and internal consistency of the translated MMQ. The instrument undergoes modifications based on a confirmatory factor analysis of the original structure, resulting in the elimination of items with cross-loadings and improvement in model fit. The modified scale is then analyzed, demonstrating strong psychometric properties. Convergent evidence is assessed by correlating MMQ subscales with the Interpersonal Reactivity Index (IRI) and Empathy Quotient (EQ), while divergent evidence is assessed by correlating aggressive traits using the Buss-Perry Aggression Questionnaire (BPAQ). The study also explores gender differences and age. Results reveal positive correlations between good mentalizing and empathy, particularly cognitive empathy, supporting the significance of positive mentalization in empathy. Negative mentalization is associated with difficulties in perspective-taking and social skills, as well as aggressive traits. Gender differences in mentalizing capacities are observed, and negative aspects of mentalization decrease with age. The availability of the Spanish translation of the MMQ, the first self-reporting scale measuring mentalization adapted to Chilean population, contributes to research aiming to understand its relationship with other psychological phenomena in different cultural context and facilitating clinical interventions with different population groups. We therefore encourage further investigation into cultural, gender and age differences in mentalization.

the reported results is available at https://anonymous.4open.science/r/MMQSpanishAdaptation-734B/README.md.

**Funding:** This research was mainly funded by the Fondecyt-Chile Grant (1200655) from the Agencia Nacional de Investigación y Desarrollo de Chile (ANID) to Edmundo Kronmüller. Additional support was provided by Fondecyt-Chile Grant (11190245) from the Agencia Nacional de In-vestigación y Desarrollo de Chile (ANID) to Nerea Aldunate, and by Fondecyt-Chile Grant (1221058) from the Agencia Nacional de Investigación y Desarrollo de Chile (ANID) to Pablo López-Silva. Funding from ANID/PIA/Basal Funds for Centers of Excellence Project FB0003 is also gratefully acknowledged. The funders had no role in study design, data collection and analysis, decision to publish, or preparation of the manuscript.

**Competing interests:** The authors have declared that no competing interests exist.

## Introduction

*Mentalizing* is the process of making sense of our own and others' behavior in terms of subjective states and mental processes [1–3]. Mentalization belongs to a broader set of socio-cognitive abilities aiming at perceiving, interpreting, and processing social information in the environment, namely, *social cognition*. Through the generation and flexible use of mental representations about the relationship between oneself and others, mentalizing is crucial in guiding socially functional behavior [4–9]. Studied in the context of the attachment-based approach to trauma and resilience [10, among many others, 11], the capacity for mentalizing is thought to be internalized during the early stages of development through reciprocal behavioral patterns in the relationship between infants and their caregivers [12–14]. Consistently, it has been shown that different attachment patterns facilitate or hinder adult mentalization processes [3, 15, 16].

Mentalization has been linked to the development of several capacities that are relevant in everyday social interaction. Indeed, since it involves an active meaning-giving perspective-taking that allows the differentiation between one's and the other's behaviors, it plays an important role in the construction of a stable sense of self and the mutuality of the social relationships we build over time [14, 16]. Mentalization has been also positively linked to language skills, emotional control, and executive control over several cognitive abilities [17]. An important capacity that has been linked to mentalization is empathy. Mentalization involves the capacity to comprehend and infer others' thoughts and emotions from different sources, including non-verbal signals such as gestures, facial expressions, and gaze, as well as knowledge about their perspectives and beliefs [18–20]. Evidence suggests that high-level mentalization skills that require the integration of knowledge about beliefs with knowledge about emotions are related to the ability to empathize with others [21]. Thus, mentalization is not only related to understanding the current individuals' emotional states, but also to the prediction of their future emotional states as a consequence of their beliefs [22, 23].

Over the years, it has been shown that the capacity to mentalize plays an important role in mental health [2, 24, 25]. For example, it is thought to be a crucial factor in the relational difficulties and interpersonal stress characteristic of schizophrenia and antisocial and borderline personality disorders [26–28]. In this context, a distinction has been made between *implicit mentalization*, corresponding to automatic-not conscious-processing, and *explicit mentalization*, which requires conscious verbal effort [29, 30]. Research suggests that social stress in subjects with antisocial traits, personality disorders, and schizophrenia would predominantly activate implicit over explicit mentalizing [30, 31]. This abnormal pattern of activation in the mentalizing system would underlie difficulties in the general understanding of social situations prompting the type of impulsive and aggressive behaviors observed in some of these conditions [32, 33]. Consistent with these findings, previous research suggests that problems in mentalization underlie emotional dysregulation, commonly observed in the aforementioned conditions [34, 35].

### The assessment of mentalization

Despite its clinical relevance as a promoter of mental health and its relation to several other cognitive and affective abilities, developing assessment methods for mentalization capacities (for others and oneself) has been elusive. Most of the methods currently available involve clinical interviews performed by trained professionals and direct observation of behaviors that could be labeled as instances of mentalizing [36]. This imposes several practical and methodological constraints on the assessment process, making it highly complex and time-consuming, especially, when applied to large samples [27]. To avoid some of these issues, researchers have

started to use self-report-based methods focused on socio-affective abilities to assess mentalization in a reliable and straightforward way [2, 27, 37]. An example of this is *the Empathy Quotient* [EQ; 38], a 60-item questionnaire (with a 40-item shorter version) that focuses on assessing cognitive empathy, emotional reactivity, and social skills. However, besides cognitive empathy, this questionnaire does not offer valuable information about key non-affective elements of mentalization involving a higher representational component, such as, for example, beliefs about the other, expectations, and intentions.

Another example of self-report-based methods is the *Interpersonal Reactivity Index* [IRI; 39, 40], a 28-item survey focused on the assessment of empathy and answered on a 5-point Likert scale. The IRI has four subscales, each one with seven items. The first two subscales focus on the cognitive aspects of empathy. While *perspective-taking* refers to the tendency to adopt the psychological point of view of others spontaneously, *fantasy* captures subjects' tendency to transport themselves imaginatively into the feelings and actions of fictitious characters in books, movies, and plays. The other two subscales focus on affective elements of empathy. While *empathic concern* assesses other-oriented feelings of sympathy and concern for unfortunate others, *personal distress* measures self-oriented feelings of personal anxiety and unease in tense interpersonal settings [40]. Problematically, the IRI's focus on empathy does not allow for identification of the more fine-grained cognitive-representational components involved in the mentalization process. This becomes a practical problem when considering the fundamental role of these components in the currently dominant definitions of mentalization, as seen in the case of the EQ.

Recently, the *Multidimensional Mentalizing Questionnaire* [MMQ; 2] has been proposed to improve self-report-based methods by assessing mentalization in a less constrained way and, crucially, beyond its affective components. The MMQ assesses mentalization capacities in adults from an attachment theory framework [see 2, 41], and it is developed based on the *Handbook of Mentalizing in Mental Health Practice* [42]. The MMQ proposes an integrated and multilevel model of mentalizing. It consists of 33 items answered on a 5-point Likert scale, ranging from 1 'Not at all' to 5 'A great deal'. It allows for the assessment of the levels of mentalization based on the construct conceptualization covering four axes: *cognitive-affective, self-other, outside-inside, and implicit-explicit* [see 41, Appendix A]. The factor model was tested through a confirmatory factor analysis (CFA) that indicated a 6-factors structure grouped in positive–or *good mentalizing*—(*reflexivity, ego-strength, and relational attunement*) and negative factors–or *bad mentalizing*—(*relational discomfort, distrust, and emotional dyscontrol*), with a satisfactory internal consistency ($\alpha = 0.75$).

Regarding the positive factors that measure good mentalizing, *reflexivity* indicated a propensity towards self-mentalization through meta-cognition, introspection, and critical thinking (e.g., 'I often try to explain what it is happening to me', "*Provo spresso a darmi delle spiegazioni su ciò che mi accade*", $\alpha = 0.89$). *Ego-strength* concerns the perception of being able to face everyday problems with an emotional resistance to stress and frustrations (e.g., 'I am able to cope with difficult situations', "*Sono in grado di affrontare situazioni difficili*", $\alpha = 0.81$), and *relational attunement* indicated the ability to tune with the emotional and cognitive states of others to understand their experiences (e.g., 'I can easily attune to other people's thinking', "*Riesco a sintonizzarmi facilmente sul pensiero altrui*", $\alpha = 0.82$). Regarding the negative factors, *relational discomfort* (opposite to *relational attunement*) captures a subject's interpersonal difficulties and the perception of being misunderstood and damaged by others (e.g., 'Others don't understand me', "*Gli altri non mi capiscono*", $\alpha = 0.76$). *Distrust* (opposite to *ego-strength*) refers to an attitude of closed-mindedness, distrust in relationships and a tendency to have a binary and rigid view of the world (e.g., 'It's better to beware of others', "*È meglio stare attenti agli altri*", $\alpha = 0.74$). Finally, *emotional dyscontrol* (opposite to *reflexivity*) refers to the

difficulty in managing own affective states and to a tendency to impulsiveness (e.g., 'I am an impulsive person', "*Sono una persona impulsiva*", α = 0.72). Recently, the psychometric properties of the MMQ have been backed up by the use of network and factorial analysis supporting the usefulness of this self-report-based method and its application in both research and clinical practice [41].

## The present study

There are more than 485 million native speakers of Spanish, making it the 3rd most spoken language in the world [43]. However, no method for the specific assessment of mentalization capacities has been developed or adapted to the Spanish language, thus creating important epistemic and clinical asymmetries for developing research about psychological promoters of mental health in that community. Indeed, as mentioned above, mentalization is associated with mental health at the level of relational difficulties and interpersonal stress in schizophrenia and antisocial and borderline personality disorders. Moreover, suicidal behavior could also be linked to a low capacity for mentalization [44, 45]. Thus, early detection of low levels of mentalization could serve both prevention and management in therapeutic interventions such as Mentalization-Based Therapy [46–48]. In this regard, it has also been proposed that mentalization is one of the primary tools for establishing therapeutic settings that could lead to a good alliance between therapist and patient. Finally, the MMQ could also be applied to therapists in training, serving to identify shortcomings and recommend the development of mentalizing skills [47, 49, 50].

Although the EQ and IRI have Spanish versions, it is essential to recall that such scales focus on assessing empathy rather than mentalizing capacities. In addition, neither the EQ nor the IR provide clear identification of cognitive components involved in the mentalization process, such as beliefs about the other, expectations, and intentions, among many others. Considering this gap in the literature, the present study examines the psychometric properties and internal consistency of the first translation of the MMQ into the Spanish. The current data was obtained from a large sample of native Spanish speakers in Chile.

Additionally, the study will aim at testing the following two hypotheses: First, since both mentalizing and empathy involve understanding and interpreting the mental and emotional states of others to facilitate social cognition and mutual understanding, we expect them to be positively associated, especially in the scales measuring their cognitive component. Second, considering that difficulties in the general understanding of social situations can lead to misinterpretations of others' intentions and behaviors, leading in turn to a higher propensity for aggression, we expect difficulties in mentalizing to be positively associated with engagement in aggressive behaviors. To tests these hypotheses, we correlated the MMQ with the Spanish versions of the EQ and IR, and the Chilean version of the *Buss and Perry Aggression Questionnaire* [BPAQ; 51], an instrument designed for the assessment of different aggressive behaviors. Thus, the study provides convergent and discriminant evidence of the validity of the MMQ.

## Methods

### Participants

419 adults participated in the study. 364 completed all the MMQ ($M_{age}$ = 29.32, SD = 11.73) participated in the study. Female participants represented 67.3% of the sample, 29.7% were males, and 3% other gender or prefer not to say. Half of the participants had completed High School (50.8%) and the other half had a college degree or equivalent (49,2%). They participated in exchange of the participation in a raffle for three gift cards of 50.000 CLP. All participants gave informed consent approved by the ethical committee of the Pontificia Universidad

Católica de Chile by checking the box corresponding to "I accept to participate" (in Spanish: *Acepto Participar*) in the web form. Data was collected from April 29[th] to June 16[th] of 2022.

## Measures

**The Empathy Quotient Scale (EQ).**    The EQ was originally developed by Baron-Cohen and Wheelwright [38] to measure empathy and showed a good internal consistency (α = 0.92). The scale is comprised of 60 items, 40 of which are statements related to empathy, either affective or cognitive (e.g., 'Other people tell me I am good at understanding how they are feeling and what they are thinking'), while the other 20 items are fillers (e.g., 'I prefer animals to humans'). Participants must answer on a 4-point Likert scale, ranging from strongly agree (1) to strongly disagree (4). According to Lawrence, Shaw [52], the EQ has a 3-factor composition. The first obtained factor is Cognitive Empathy (CE), with items related to the capacity to attribute mental states, representing affective states (e.g., 'I can tell if someone is masking their true emotion'), epistemic states (e.g., 'I find it easy to put myself in somebody else's shoes') and desires (e.g., 'I can easily work out what another person might want to talk about'). The second factor is Emotional Reactivity (ER) which measures the tendency to react emotionally in response to others' mental states (e.g., 'seeing people cry doesn't really upset me'). Finally, the third factor is Cognitive Skills (CS), which asses the spontaneous use of empathic skills (e.g., 'I often find it difficult to judge whether something is rude or polite'). The overall Cronbach's alpha for the Chilean version was 0.83 [53], and in this study was 0.78.

**The Interpersonal Reactivity Index (IRI).**    The IRI measures empathy in four dimensions: emphatic concern, perspective-taking, personal distress, and fantasy. The original English version [39, 40] shows good internal consistency with Cronbach's alphas ranging from 0.71 to 0.77. As defined by Davis [40], emphatic concern (EC) evaluates the tendency to sympathize with others who suffer (e.g., 'I often have tender, concerned feelings for people less fortunate than me'). Perspective-taking (PT) refers to the tendency to take the psychological point of view of others (e.g., 'I try to look at everybody's side of a disagreement before I make a decision'). Fantasy (FT) assesses the capacity to imaginatively feel and think like fictional characters in movies and books (e.g., 'I really get involved with the feelings of the characters in a novel'). Finally, Personal distress (PD) measures the level of anxiety produced by adverse scenarios (e.g., 'I sometimes feel helpless when I am in the middle of a very emotional situation'). The instrument presents statements on a 5 points Likert scale, ranging from 1 (does not describe me well) to 5 (describes me very well). Its psychometric structure was tested in a Chilean sample [54], showing a consistent pattern with the original (α = 0.76). For the current study, the observed Cronbach's alpha was 0,79.

**Buss and Perry Aggression Questionnaire (BPAQ).**    The original BPAQ [55] measures four forms of aggression through self-report, with statements such as 'Sometimes I can't control the impulse to hit another person'. Participants must indicate on a 5-point Likert scale, ranging from 1 (extremely uncharacteristic of me) to 5 (extremely characteristic of me). Its psychometric structure was tested with a sample of Chilean students [51], showing Cronbach's alphas ranging from 0.60 to 0.80. The questionnaire shows a composition with four factors. Physical aggression and verbal aggression asses the activation of aggressive behaviors. Whereas hostility reflects the cognitive and anger the affective aspects of aggressive traits. In this study, BPAQ showed a good internal consistency (α = 0.88).

**Items translation and adaptation.**    The process of translating and adapting the items focused on reducing irrelevant features of the construct by adapting them to the cultural characteristics of the target population [American Psychological 56]. For this reason, all items were translated in two ways: a literal translation and an adapted translation, aiming for the latter to

address the expressive modes specific to Chilean cultural and linguistic context. Both translations and the original English items were presented to an expert panel composed of two psycholinguists and two linguists. The panel members analyzed and evaluated the translations following a process adapted from Solano-Flores, Contreras-Niño [57]. This process examined ten dimensions in which adaptation errors can occur during translation: style, format, conventions, information, grammar, semantics, register, culture, and origin (error in the original item). Items in which errors were detected were discussed until a consensus was reached regarding an error-free final version.

Five cognitive interviews (3 women) with individuals without formal education in psychology were conducted. The procedure consisted of two stages. In the first stage, participants completed the instrument, and in the second stage, they were interviewed for each item, answering four questions: 1) What is the item asking you to respond to? 2) What did you answer? 3) Why did you choose this response? 4) Was there anything in the item that seemed unclear or confusing to you? A total of 21 items with errors were identified and agreed upon, of which 19 were in the register dimension, 7 in the semantic area, and 1 in the information area. In all these cases, the errors were corrected, and a suitable version from the panel's perspective was reached.

## Data analyses

With the structure proposed and tested by Gori, Arcioni [2], we conducted a confirmatory factor analysis (CFA) of the adapted MMQ, based on the common factors model. In addition, we tested 3 additional models: bifactor, one-factor, and hierarchical factor model, to determine the best factor structure. We then test their reliability by computing Cronbach's alpha for each subscale. In order to contrast our initial hypotheses, we compute Pearson correlations with all scales and subscales of the different instruments for convergent and divergent evidence. Finally, following Gori, Arcioni [2], we conducted a *t-test* on the means for males and females on each subscale for all the MMQ scales. To test whether the constructs have the same meaning for those who identify themselves as men or women, we performed an analysis of invariance. Following the sequence proposed by Wu & Estabrook [58], we first compared the baseline configural model with a restricted threshold model, then the factor loadings are restricted, and the intercept is forced to be equal across groups. Finally, the error variance for both groups is set to be equal. Given the ordinal nature of the Likert scale used in the instrument, we used polychoric correlations to estimate the model's fit. This methodological approach has been widely argued to be the most suitable for modeling ordinal data [59–61]. All analyses were conducted with R Lavaan [v0.6.15; 62].

## Results

### Descriptive analyses

Table 1 presents each item along with its respective mean, median, and standard deviation. The original factorial structure is shown in Table 2. Based on the item means, a tendency to respond positively (that applies to one's life) can be observed when the content reflects a socially valued attitude. Thus, statements related to self-reflection on one's behaviors and emotions and the importance of understanding and empathizing with others have a higher average than statements that indicate a negative attitude towards social relationships or a self-perception of low competence in self-control. A tendency toward positive response bias can affect the instrument's sensitivity by reducing variance. However, based on the results in Table 1, it can be concluded that participants did utilize the full scale from one to five.

**Table 1. Descriptive statistics.**

| Item | mean | sd | median |
|---|---|---|---|
| 1. Frecuentemente intento explicar lo que me está pasando *(I often try to explain what is happening to me)* | 3.67 | 1.05 | 4.00 |
| 2. Soy una persona impulsiva *(I am an impulsive person)* | 2.66 | 1.09 | 2.00 |
| 3. A veces experimento cambios de ánimo que no puedo controlar *(I sometimes experience mood swings I can't control)* | 3.22 | 1.18 | 3.00 |
| 4. Soy capaz de captar los aspectos más profundos de las personas que me rodean *(I'm able to get the deepest aspects of people around me)* | 3.78 | 0.86 | 4.00 |
| 5. Me puedo conectar con el estado mental de los demás *(I can tune in other people's mental states)* | 3.72 | 0.87 | 4.00 |
| 6. Para entender las acciones de los demás, es fundamental comprender lo que sienten *(Understanding what others feel is crucial in understanding their actions)* | 4.50 | 0.70 | 5.00 |
| 7. A veces siento que estoy perdiendo el control de mis emociones *(I sometimes feel like I am losing control of my emotions)* | 2.89 | 1.21 | 3.00 |
| 8. Soy capaz de reflexionar sobre mis acciones *(I am able to reflect on my behaviours)* | 4.55 | 0.62 | 5.00 |
| 9. Relacionarme con otras personas me impide ser yo mismo/a *(Relationships with other people prevent me from being myself)* | 2.06 | 1.00 | 2.00 |
| 10. Me interesa entender cómo funciona mi mente *(I'm interested in understanding my mental processes)* | 4.70 | 0.58 | 5.00 |
| 11. Puedo tolerar las frustraciones de la vida cotidiana *(I can tolerate frustrations of daily life)* | 3.69 | 0.88 | 4.00 |
| 12. Las personas no me entienden *(Others don't understand me)* | 2.56 | 0.96 | 2.00 |
| 13. Es mejor tener cuidado con los demás *(It's better to beware of others)* | 3.18 | 1.02 | 3.00 |
| 14. Soy capaz de empatizar con otros cuando me cuentan algo *(I'm able to empathize with others when they tell me something)* | 4.44 | 0.58 | 4.00 |
| 15. Me asusta abrirme con los demás *( I am afraid to open up with other people)* | 3.06 | 1.16 | 3.00 |
| 16. Reflexiono sobre lo que me pasa *(I ponder over what happens to me)* | 4.47 | 0.61 | 5.00 |
| 17. Encuentro útil analizar mi comportamiento *(I find beneficial to analyse my behaviour)* | 4.52 | 0.65 | 5.00 |
| 18. Suelo preguntarme por qué pasan las cosas *(I often think about why things happen)* | 4.26 | 0.86 | 4.00 |
| 19. Para mí las cosas son blancas o son negras *(For me things are either white or black)* | 1.91 | 1.11 | 2.00 |
| 20. No confío en los demás *(I don't trust others)* | 2.43 | 1.05 | 2.00 |
| 21. Soy sensible a lo que le pasa a los demás *(I am sensitive to what happens to others)* | 4.02 | 0.83 | 4.00 |
| 22. Generalmente, puedo adaptarme a diferentes contextos sin dificultad *(I can usually adapt myself to different contexts with no difficulties)* | 3.68 | 0.90 | 4.00 |
| 23. A veces tengo emociones contradictorias *(It happens to me to have conflicting emotions)* | 3.65 | 1.01 | 4.00 |
| 24. Soy capaz de resolver problemas difíciles cuando se me presentan en la vida *(I am able to sort out difficult problems when life presents those to me)* | 3.95 | 0.73 | 4.00 |
| 25. Soy capaz de sobrellevar la carga emocional de situaciones estresantes *(I am able to bear the emotional load of stressful situations)* | 3.70 | 0.93 | 4.00 |
| 26. Cuando siento una emoción intensa, puedo controlarla *(When I feel an intense emotion, I can control it)* | 3.48 | 0.92 | 4.00 |
| 27. La gente me abandona *(People abandon me)* | 2.24 | 1.09 | 2.00 |
| 28. Puedo conectarme fácilmente con lo que piensan las otras personas *(I can easily attune to other people's thinking)* | 3.80 | 0.69 | 4.00 |
| 29. Es mejor tener cuidado con los desconocidos *(It's better to beware of strangers)* | 3.58 | 1.00 | 4.00 |
| 30. Soy capaz de enfrentar situaciones difíciles *(I am able to cope with difficult situations)* | 4.05 | 0.68 | 4.00 |
| 31. Soy una persona que piensa en los demás *(I am a thoughtful person)* | 4.25 | 0.64 | 4.00 |
| 32. Me gusta entender por qué ciertas cosas me pasan a mí *(I'm keen on understanding why certain things happen to me)* | 4.09 | 0.90 | 4.00 |
| 33. Algunas personas son la causa de mis problemas *(Some people are the cause of my problems)* | 2.59 | 1.11 | 2.00 |

**Table 2. Original factor structure of the Multidimensional Mentalizing Questionnaire.**

| Factor | Item |
|---|---|
| Reflexivity | 1 |
| | 6 |
| | 8 |
| | 10 |
| | 16 |
| | 17 |
| | 18 |
| | 31 |
| | 32 |
| Ego Strength | 11 |
| | 22 |
| | 24 |
| | 25 |
| | 26 |
| | 30 |
| Relational Attunement | 4 |
| | 5 |
| | 14 |
| | 21 |
| | 28 |
| Relational Discomfort | 9 |
| | 12 |
| | 15 |
| | 27 |
| | 33 |
| Distrust | 13 |
| | 19 |
| | 20 |
| | 29 |
| Emotional Dyscontrol | 2 |
| | 3 |
| | 7 |
| | 23 |

## Confirmatory factor analysis

We carried out a confirmatory factor analysis for the factorial structure proposed by Gori, Arcioni [2], as presented in Table 2. As can be seen in Table 3, the fit indicators and explained variance are below current standards [63]. Aiming to improve the fit and, at the same time, safeguard the simplicity and the parsimony of the model, we look for items that correlated with one or more additional factors different than the one to which they were initially associated based on theoretical and empirical grounds. Specifically, we examined the modification indices provided by the Lavaan package v. 0.6.15 [62] to identify items to which adding an extra parameter, i.e., a factor weight, substantially improved the fit indicators of the model. Following this rationale, five items exhibited substantial cross-loadings (18, 26, 30, 31, and 32). Three of these items originally belonged to the Reflexivity subscale, which had the largest number of items, and the remaining two belonged to Ego Strength, the second largest subscale. As

**Table 3. Fit indices of the original and modified models.**

| Model | $\chi^2$ | df | $\chi^2/df$ | p | CFI | TLI | RMSEA | SRMR |
|---|---|---|---|---|---|---|---|---|
| Original Common Factors Model | 1,660.23 | 480 | 3.46 | < .001 | 0.94 | 0.93 | 0.08 | 0.09 |
| Modified Common Factors Model | 836.90 | 335 | 2.50 | < .001 | 0.96 | 0.95 | 0.06 | 0.08 |
| One factor model | 4649.83 | 350 | 13.28 | < .001 | 0.62 | 0.59 | 0.18 | 0.16 |
| Bifactor model | 1460.04 | 322 | 4.53 | < .001 | 0.90 | 0.88 | 0.09 | 0.10 |
| Hierarchical factor model | 1664.92 | 344 | 4.83 | < .001 | 0.88 | 0.87 | 0.10 | 0.11 |

shown in Table 3, the model's fit indicators improved substantially by removing those items, placing them within the accepted psychometric standards for the common factors model [63]. The modified model also performed better than three alternative models: one factor, bifactor and hierarchical.

Regarding the factor loadings (see Table 4), only two items in the reflexivity scale show loadings below 0.4. When comparing these factor weights with those of the original scale reported by Gori, Arcioni [2], it can be observed that the range of these values is not significantly different from those obtained in the version reported here, falling within an appropriate range. Similarly, the reliability indices for each are within an acceptable range (see Table 5).

**Table 4. Factor loading (lambda).**

| Item | Reflexivity | Ego Strength | Relational Attunement | Relational Discomfort | Distrust | Emotional Dyscontrol |
|---|---|---|---|---|---|---|
| 1 | 0.377 | | | | | |
| 6 | 0.299 | | | | | |
| 8 | 0.684 | | | | | |
| 10 | 0.648 | | | | | |
| 16 | 0.855 | | | | | |
| 17 | 0.739 | | | | | |
| 11 | | 0.723 | | | | |
| 22 | | 0.608 | | | | |
| 24 | | 0.819 | | | | |
| 25 | | 0.792 | | | | |
| 4 | | | 0.768 | | | |
| 5 | | | 0.748 | | | |
| 14 | | | 0.658 | | | |
| 21 | | | 0.520 | | | |
| 28 | | | 0.727 | | | |
| 9 | | | | 0.573 | | |
| 12 | | | | 0.700 | | |
| 15 | | | | 0.539 | | |
| 27 | | | | 0.700 | | |
| 33 | | | | 0.553 | | |
| 13 | | | | | 0.798 | |
| 19 | | | | | 0.421 | |
| 20 | | | | | 0.733 | |
| 29 | | | | | 0.723 | |
| 2 | | | | | | 0.495 |
| 3 | | | | | | 0.823 |
| 7 | | | | | | 0.880 |
| 23 | | | | | | 0.645 |

**Table 5. Descriptive statistics for each subscale.**

| Subscales | n | mean | sd | median | min | max | range | shapiro | α |
|---|---|---|---|---|---|---|---|---|---|
| Reflexivity | 364 | 4.40 | 0.41 | 4.50 | 2.67 | 5.00 | 2.33 | 0.00 | 0.63 |
| Ego-Strength | 364 | 3.75 | 0.66 | 3.75 | 1.25 | 5.00 | 3.75 | 0.00 | 0.77 |
| Relational Attunement | 364 | 3.95 | 0.54 | 4.00 | 1.80 | 5.00 | 3.20 | 0.00 | 0.74 |
| Relational Discomfort | 364 | 2.50 | 0.71 | 2.40 | 1.00 | 5.00 | 4.00 | 0.00 | 0.70 |
| Distrust | 364 | 2.77 | 0.75 | 2.75 | 1.00 | 5.00 | 4.00 | 0.00 | 0.69 |
| Emotional Dyscontrol | 364 | 3.11 | 0.86 | 3.25 | 1.00 | 5.00 | 4.00 | 0.00 | 0.76 |
| **Positive Mentalizing** | 364 | 4.04 | 0.38 | 4.05 | 2.69 | 5.00 | 2.31 | 0.02 | 0.77 |
| **Negative Mentalizing** | 364 | 2.80 | 0.61 | 2.78 | 1.08 | 4.85 | 3.77 | 0.42 | 0.83 |

While these values might be considered relatively low, the small number of items per scale can account for them, as Cronbach's Alpha is sensitive to the number of items on the scale being tested.

Finally, as seen in Table 6, the structure of correlations between the subscales is consistent with the underlying constructs. This can be primarily seen in the pattern of positive and negative correlations between the subscales related to good mentalization (Reflexivity, Ego Strength, and Relational Attunement) and those related to poor mentalization (Relational Discomfort, Distrust, and Emotional Dyscontrol). For example, Relational Discomfort has a strong positive correlation with Distrust ($r = .70$), contrasting with the negative correlation with Ego Strength ($r = -.60$). While moderate and strong correlations can be observed between some of the subscales, in general, correlations among them are low or non-existent, indicating a low level of redundancy in the overall scale.

## Convergent and divergent evidence

According to our predictions, higher scores on the positive MMQ are strongly associated with higher scores on the EQ ($r = 0.53$; $p < 0.01$), especially in the case of the cognitive empathy subscale ($r = 0.56$; $p < 0.01$). Although a small correlation was found with the complete IRI ($r = 0.14$; $p > 0.05$), it was observed that the positive MMQ has a moderate positive correlation with the perspective-taking ($r = 0.34$; $p < 0.01$) and empathic concern ($r = 0.32$; $p < 0.01$) subscales.

**Table 6. Correlations between MMQ subscales.**

| Factor | 1 | 2 | 3 | 4 | 5 |
|---|---|---|---|---|---|
| 1. Reflexivity | | | | | |
| 2. Ego-Strength | .21** <br> *.39** | | | | |
| 3. Relational Attunement | .38** <br> *.51** | .16** <br> *.26** | | | |
| 4. Relational Discomfort | -.08 <br> *-.12 | -.45** <br> *-.60** | -.05 <br> *-.09 | | |
| 5. Distrust | .01 <br> *.04 | -.26** <br> *-.32** | -.10 <br> *-.11 | .53** <br> *.70** | |
| 6. Emotional Dyscontrol | .10 <br> *.11 | -.39** <br> *-.51** | .17** <br> *.20=** | .48** <br> *.64** | .33** <br> *.40** |

Note.

* indicates $p < .05$

** indicates $p < .01$. Latent scores correlation shown in italics.

**Table 7. Correlations between MMQ, EQ, and IRI.**

| Variable | Reflexivity | Ego-Strength | Relational Attunement | Relational Discomfort | Distrust | Emotional Dyscontrol |
|---|---|---|---|---|---|---|
| 1. Cognitive Empathy | .30** | .27** | .63** | -.14* | -.07 | .05 |
| | .34** | .32** | .63** | -.16** | -.09 | .06 |
| 2. Social Skills | .15** | .29** | .21** | -.26** | -.20** | -.23** |
| | .17* | .31** | .26** | -.30** | -.20** | -.24** |
| 3. Emotional Reactivity | .24** | .06 | .43** | -.30** | -.31** | -.05 |
| | .22** | .07 | .65** | -.35** | -.34** | -.04 |
| 4. Fantasy | .23** | -.15** | .21** | .13* | .01 | .23** |
| | .19** | -.16* | .23** | .15* | .04 | .28** |
| 5. Personal Distress | -.06 | -.52** | -.04 | .32** | .24** | .32** |
| | -.11 | -.59** | .00 | .37** | .36** | .35** |
| 6. Perspective Taking | .27** | .20** | .28** | -.18** | -.28** | -.16** |
| | .27** | .21** | .38** | -.21** | -.26** | -.17** |
| 7. Empathic Concern | .25** | .01 | .49** | -.11 | -.22** | .07 |
| | .21** | .00 | .74** | -.13* | -.26 | .09 |
| 8. Physical Aggression | -.04 | -.16** | -.23** | .40** | .37** | .30** |
| | -.02 | -.16** | -.31** | .46** | .40** | .33** |
| 9. Verbal Aggression | .04 | -.03 | -.04 | .25** | .23** | .25** |
| | .11 | -.02 | -.04 | .28** | .25** | .26** |
| 10. Anger | .02 | -.20** | .03 | .30** | .30** | .55** |
| | .01 | -.22 | .02 | .34** | .29** | .60** |
| 11. Hostility | -.04 | -.38** | -.10 | .65** | .51** | .49** |
| | -.03 | -.41** | -.18** | .75** | .56** | .56** |
| 12. Positive MMQ | .66** | .73** | .70** | -.32** | -.20** | -.11* |
| 13. Negative MMQ | .02 | -.46** | .02 | .83** | .77** | .79** |
| 15. EQ | .31** | .26** | .57** | -.30** | -.25** | -.08 |
| | .32** | .30** | .69** | -.35** | -.27** | -.08 |
| 16. IRI | .27** | -.22** | .37** | .09 | -.07 | .21** |
| | .23** | -.25** | .52** | .01 | -.07 | .27** |
| 17. BPAQ | -.01 | -.27** | -.10 | .53** | .46** | .53** |
| | .02 | -.28** | -.16* | .61** | .50** | .59** |

Note.

* indicates $p < .05$

** indicates $p < .01$. Correlations between latent scores of the subscales of MMQ and the raw scores of the other scales in *italics*.

The negative MMQ only shows a weak negative correlation with the EQ ($r$ = -0.26; $p < 0.01$), especially with social skills ($r$ = -0.29; $p < 0.01$) and emotional reactivity ($r$ = -0.27; $p < 0.01$). Among the IRI subscales, the negative MMQ had a weak negative correlation with perspective taking ($r$ = -0.25; $p < 0.01$), but moderate positive correlation with personal distress ($r$ = 0.36; $p < 0.01$). Low negative correlations are observed regarding the relationship between the negative MMQ and empathy scales.

As can be seen in Table 7, regarding the subscales of the positive MMQ, it can be observed that the strongest positive correlations were found between relational attunement of the positive MMQ and cognitive empathy of the EQ ($r$ = 0.63; $p < 0.01$), as well as with empathic concern of the IRI ($r$ = 0.72; $p < 0.01$). Additionally, there is a strong negative correlation between ego-strength of the positive MMQ and personal distress of the IRI ($r$ = -0.59; $p < 0.01$).

**Table 8. Invariance analysis.**

| Restriction | df | χ2 | Δ χ2 | Δ df | p-value | CFI | TLI | RMSEA | SRMR |
|---|---|---|---|---|---|---|---|---|---|
| Unrestricted | 388 | 691.33 | | | | | | | |
| Threshold | 430 | 716.84 | 48.21 | 42 | .235 | .96 | .95 | .06 | .08 |
| Thresholds and loadings | 446 | 748.16 | 20.71 | 16 | .189 | .96 | .96 | .06 | .08 |
| Thresholds, loadings and intercepts | 462 | 771.73 | 20.90 | 16 | .182 | .96 | .96 | .06 | .08 |
| Thresholds, loadings, intercepts, and errors | 484 | 797.48 | 19.55 | 22 | .611 | .96 | .96 | .06 | .08 |

The MMQ also showed correlations with the BPAQ. A weak negative relationship was observed between the positive MMQ and the BPAQ ($r = -0.20$; $p < 0.01$). As can be seen in Table 7, the highest correlation was obtained between ego-strength and hostility ($r = -0.41$; $p < 0.01$). Strong correlations were found between the negative MMQ and the BPAQ ($r = 0.64$; $p < 0.01$), particularly with hostility ($r = 0.68$; $p < 0.01$). Hostility showed moderate to strong correlations with the subscales of relational discomfort ($r = 0.75$; $p < 0.01$), distrust ($r = 0.56$; $p < 0.01$), and emotional dyscontrol ($r = 0.56$; $p < 0.01$). Additionally, anger also positively correlated with emotional dyscontrol ($r = 0.60$; $p < 0.01$).

### Further analyses: Gender differences and correlation with age

As it can be seen in Table 8, invariance analysis shows no difference between male and female. Comparisons between both groups show that women scored higher in their self-perception of good mentalizing and empathy than men. Similarly, women exhibit a lower tendency towards aggression, as measured by the BPAQ (see Table 9). In addition, we found an inverse association between the negative aspects of mentalization subscales and age. In other words, negative aspects of mentalization tend to decrease over time. The same tendency was observed in aggressive traits (see Table 10).

### Discussion

This paper examines psychometric properties and internal consistency of the first translation and adaptation of the MMQ into the Spanish, using a native Spanish-speaking sample in Chile. We first conducted a confirmatory factor analysis based on psychometric structure of the original test [reported in 2]. We then modify the instrument by eliminating the items showing cross-loadings with two or more factors and where adding the corresponding parameters would improve the model's fit. A second factor analysis on the modified scale provide evidence of a good psychometric performance of the shorter and more parsimonious scale. To provide convergent and divergent evidence, we correlated the different subscales of the MMQ with all the subscales of the IRI and EQ, since both asses, indirectly, some components of mentalization. Also, we include an instrument measuring the tendency for aggressive behaviors, looking for evidence of a relationship between the ability to mentalize and the engagement in these types of behaviors. Finally, we explore gender differences and the association of the MMQ with age.

Our results provide evidence of consistency between the MMQ and other scales that indirectly measure the capacity for mentalization. Supporting our first hypothesis, our results show that positive mentalizing correlates with cognitive components of empathy, such as cognitive empathy as measured in the EQ and perspective-taking in the IRI. This reaffirms that positive mentalization is a fundamental component of empathy, and training mentalization could positively impact the development of better empathetic skills. The subscale of relational attunement shows strong positive correlations with cognitive empathy in the EQ and empathic

**Table 9. Gender differences.**

| Scale | Total | | | Women | | | Men | | | | | |
|---|---|---|---|---|---|---|---|---|---|---|---|---|
| | N | Mean | SD | N | Mean | SD | N | Mean | SD | t | df | p |
| **MMQ** | | | | | | | | | | | | |
| Reflexivity | 353 | 4.40 | 0.41 | 245.00 | 4.43 | 0.40 | 108.00 | 4.34 | 0.45 | -1.86 | 183.99 | 0.07 |
| Ego-Strength | 353 | 3.77 | 0.65 | 245.00 | 3.74 | 0.66 | 108.00 | 3.84 | 0.63 | 1.27 | 213.50 | 0.21 |
| Relational Attunement | 353 | 3.95 | 0.54 | 245.00 | 4.06 | 0.51 | 108.00 | 3.71 | 0.55 | -5.51 | 189.72 | 0.00 |
| Relational Discomfort | 353 | 2.49 | 0.72 | 245.00 | 2.49 | 0.69 | 108.00 | 2.50 | 0.77 | 0.17 | 186.46 | 0.87 |
| Distrust | 353 | 2.77 | 0.75 | 245.00 | 2.79 | 0.74 | 108.00 | 2.74 | 0.78 | -0.57 | 195.50 | 0.57 |
| Emotional Dyscontrol | 353 | 3.10 | 0.86 | 245.00 | 3.19 | 0.85 | 108.00 | 2.88 | 0.87 | -3.09 | 201.12 | 0.00 |
| MMQ Positive | 353 | 4.04 | 0.38 | 245.00 | 4.08 | 0.37 | 108.00 | 3.96 | 0.38 | -2.59 | 201.49 | 0.01 |
| MMQ Negative | 353 | 2.79 | 0.62 | 245.00 | 2.82 | 0.62 | 108.00 | 2.71 | 0.61 | -1.61 | 205.79 | 0.11 |
| **EQ** | | | | | | | | | | | | |
| Cognitive Empathy | 298 | 1.12 | 0.40 | 208.00 | 1.16 | 0.38 | 90.00 | 1.03 | 0.41 | -2.71 | 158.02 | 0.01 |
| Social Skills | 298 | 1.06 | 0.31 | 208.00 | 1.09 | 0.30 | 90.00 | 0.98 | 0.32 | -2.89 | 157.60 | 0.00 |
| Emotional Reactivity | 298 | 1.28 | 0.40 | 208.00 | 1.37 | 0.38 | 90.00 | 1.08 | 0.39 | -5.89 | 163.79 | 0.00 |
| EQ Total | 298 | 1.15 | 0.28 | 208.00 | 1.21 | 0.26 | 90.00 | 1.03 | 0.29 | -5.01 | 153.43 | 0.00 |
| **IRI** | | | | | | | | | | | | |
| Empathic Concern | 285 | 2.92 | 0.73 | 200.00 | 3.04 | 0.66 | 85.00 | 2.66 | 0.81 | -3.83 | 133.65 | 0.00 |
| Fantasy | 285 | 2.69 | 0.91 | 200.00 | 2.84 | 0.89 | 85.00 | 2.36 | 0.87 | -4.20 | 161.77 | 0.00 |
| Personal Distress | 285 | 1.67 | 0.81 | 200.00 | 1.73 | 0.80 | 85.00 | 1.53 | 0.82 | -1.94 | 155.20 | 0.05 |
| Perspective Taking | 285 | 2.92 | 0.67 | 200.00 | 2.92 | 0.68 | 85.00 | 2.92 | 0.65 | -0.04 | 166.67 | 0.96 |
| IRI Total | 285 | 2.55 | 0.48 | 200.00 | 2.63 | 0.45 | 85.00 | 2.36 | 0.51 | -4.20 | 141.78 | 0.00 |
| **BPAQ** | | | | | | | | | | | | |
| Physical Aggression | 279 | 1.54 | 0.56 | 196.00 | 1.44 | 0.50 | 83.00 | 1.77 | 0.62 | 4.25 | 127.67 | 0.00 |
| Verbal Aggression | 279 | 2.61 | 0.63 | 196.00 | 2.53 | 0.61 | 83.00 | 2.81 | 0.64 | 3.38 | 147.62 | 0.00 |
| Hostility | 279 | 2.30 | 0.72 | 196.00 | 2.27 | 0.69 | 83.00 | 2.36 | 0.79 | 0.87 | 137.11 | 0.39 |
| Anger | 279 | 2.40 | 0.69 | 196.00 | 2.40 | 0.69 | 83.00 | 2.39 | 0.70 | -0.12 | 151.86 | 0.91 |
| BPAQ Total | 279 | 2.21 | 0.50 | 196.00 | 2.16 | 0.48 | 83.00 | 2.33 | 0.52 | 2.57 | 143.57 | 0.01 |

concern in the IRI. Cognitive empathy is a component of empathy that involves representing the cognitive aspects of someone else's emotional states, such as recognizing their affective beliefs, thoughts, or intentions [64]. The observed correlation reflects shared aspects between mentalizing and cognitive empathy, which involve their capacity to understand the other's mental states by inferring their thoughts, beliefs, and emotions [22, 65]. Empathic concern refers to emotional responses such as compassion, sympathy, or care for others experiencing suffering or distress. Here, mentalization implies inferring the emotional experiences of others. This inference induces a sense of concern and empathy for the individual's well-being, promoting social behaviors such as prosociality [66–68]. Relational attunement plays a special role in cognitive empathy and empathic concern. This latter dimension refers to the capacity to connect or align with someone else's emotions, needs, or beliefs. By attuning to others and understanding their emotions and experiences, individuals can improve a cognitive understanding of other's mental states and establish emotional connections with others. Additionally, an inverse relation between ego-strength and personal distress (IRI) was found. Ego-strength refers to an individual's capacity to face adversity and cope with stressors and problems through self-regulation [69–71]. Importantly, people with a decreased capacity to manage and regulate their emotions adaptively appears to suffer more personal distress [72, 73], which denotes the negative emotional experience in response to these stressors [74].

**Table 10. Correlation with age.**

| Scale | Age |
|---|---|
| Reflexivity | -.10 |
| Ego-Strength | .27** |
| Relational Attunement | .03 |
| Relational Discomfort | -.34** |
| Distrust | -.24** |
| Emotional Dyscontrol | -.33** |
| Cognitive Empathy | .03 |
| Social Skills | .05 |
| Emotional Reactivity | .09 |
| Fantasy | -.26** |
| Personal Distress | -.17** |
| Perspective Taking | -.03 |
| Empathic Concern | .04 |
| Physical Aggression | -.18** |
| Verbal Aggression | -.03 |
| Rage | -.07 |
| Hostility | -.28** |
| **MMQ Positive** | .13* |
| **MMQ Negative** | -.38** |
| **EQ Total** | .08 |
| **IRI Total** | -.19** |
| **BPAQ Total** | -.19** |

Note.

\* indicates $p < .05$.

\*\* indicates $p < .01$.

Regarding our second hypothesis, we observed that personal distress and emotional reactivity positively correlate with bad mentalizing (negative MMQ). Bad mentalizing is measured through relational discomfort, distrust, and emotional dyscontrol, and it is the opposite pole of positive mentalizing. Negative mentalizing refers to failures and distortions in mentalizing, and it is related to problems in connecting with others. This might be produced by an increased emotional reactivity and impulsiveness, attitudes of closed-mindedness, distrust in relationships, and relational insecurity caused by the perception of being misunderstood [2]. This study shows that poor mentalizing is closely related to problems with perspective-taking and social skills. Previous literature established a relationship between poor mentalizing and aggressive traits [32]. Notably, a strong relation between mentalizing and aggressive traits was found in this study, especially in the case of bad mentalizing and hostility. We observed hostility related to relational discomfort, distrust, and emotional dyscontrol. Furthermore, emotional dyscontrol was correlated with anger. These results are in line with the obtained by Gori, Arcioni [2] where bad mentalizing was related to impulsiveness. These authors and others suggest that a self-regulation deficiency would underlie impulsive and negative reactions [2, 75]. Low ego-strength with high relational discomfort and distrust in relationships could lead to misunderstandings of the intentions and motivations of others. These misunderstandings could conduct to impulsivity and hostility as a primal adaptive response. In this sense, people may respond defensively or more aggressively due to increased discomfort or lack of trust [51, 76]. Additionally, deficiencies in emotional control are related to hostility and anger,

since people with difficulties regulating their emotions may be more prone to be aggressive and hostile [77–79].

Our analysis found gender differences in mentalizing capacities. Previous studies have shown gender differences where women score higher than men in empathy measures [52, 80]. However, those scales are not specifically designed to measure mentalization capacities. In our study, women exhibit higher levels of reflexivity and relational attunement than men, indicating positive aspects of mentalizing. However, women also report a higher self-perception of emotional dyscontrol than men, which could be related to gender differences in emotional regulation strategies [81–83]. Further research is needed to clarify the connections between these results and how cultural factors could impact the development of mentalization abilities in different genders.

Finally, we observed in an exploratory analysis that the negative aspects of mentalization measured in the MMQ tend to decrease with age. This interesting finding offers the opportunity to address mentalization from the perspective of the subjects' life cycle and developmental stage. Arguably, the decrease in negative aspects of mentalization could be explained by the way in which life experiences shape relational discomfort, distrust, and emotional dyscontrol. Integrating meaningful life experiences into understanding others would improve emotional regulation and a greater ability to navigate relationships effectively [84, 85]. It is important to note that these results are exploratory and should be interpreted with caution considering that our sample has a mean age of 29.32 years old and the distribution is positively skewed. Moreover, future work should address the generalizability of our findings across life span through invariance analysis. In sum, further research is needed to gain a more comprehensive understanding of gender differences and age-related patterns in mentalization.

It is also important to note that measuring mentalization capacity through self-reporting poses a challenge shared with other self-report instruments that measure specific capacities without directly observing performance in a task that assesses that capacity. In fact, we can argue that the MMQ measures individuals' self-perception of their mentalization abilities. It is a matter of debate and empirical investigation to determine how self-perception of competence is related to actual performance in tasks or activities that require that competence for execution. Concerning this, one possible avenue for research would be to correlate self-perception of mentalization with the judgments of others who interact with the individuals being evaluated on a daily basis. Crossed-examinations of mentalizing capacities from the 1st and 3rd person could enrich the use of self-report-based methods. Similarly, it could be valuable to associate the results of comprehensive psychological interviews and assessments with the scores obtained on the MMQ. By incorporating these approaches, researchers could gain a broader understanding of the relationship between self-perception of mentalization, external assessments, and objective performance in tasks that require mentalization abilities. This would contribute to the ongoing investigation and refinement of mentalization measurement tools.

The availability of an instrument that measures mentalization in various dimensions, acknowledging the complexity of this capacity, allows us to advance our understanding of it in clinical populations. Future research could compare MMQ's results in different clinical populations with non-clinical control samples to understand how mentalizing works in those diagnoses. With this, more specific interventions focused on training mentalizing abilities could be developed. Additionally, an instrument of this kind enables us to study potential differences and invariances of the construct based on gender and stage of the life cycle. Furthermore, it allows us to conduct transcultural research on this fundamental human capacity for sociability, which may vary to different degrees among individuals from different cultural backgrounds and environments. In conclusion, an instrument that measures mentalization, accounting for

the complexity of the construct, provides the opportunity for conducting research and clinical interventions. Moreover, making this type of instrument available in different languages allows us to explore the variations of mentalization capacity across different cultural backgrounds.

In conclusion, the Spanish adaptation of the MMQ presented in this work demonstrates robust psychometric properties in the Chilean population. The study provides evidence of how the different dimensions of mentalizing may be related to empathic abilities and aggressive behaviors. Considering that the original version was developed in Italian, it is crucial to address potential cultural differences, as they may influence the behavioral expression of mentalization and item functioning [86, 87]. Therefore, we advocate for cross-cultural studies that, among other questions, test cultural invariance. Additionally, exploring the relationship between age and mentalization and other pertinent demographic analyses in forthcoming investigations is crucial. Finally, the adaptation of the MMQ presented in this paper is the first tool that will serve for self-administered measurements of mentalization in Chile, providing a standardized and valid measure that can inform psychological treatment and research.

## Acknowledgments

Thanks to David Torres for his helpful advice regarding data analysis. We also thank our reviewer, whose comments and suggestions substantially improved the quality of our work.

## Author Contributions

**Conceptualization:** Nerea Aldunate, Cristian Brotfeld, Edmundo Kronmüller.

**Data curation:** Cristian Brotfeld, Edmundo Kronmüller.

**Formal analysis:** Cristian Brotfeld, Edmundo Kronmüller.

**Funding acquisition:** Edmundo Kronmüller.

**Methodology:** Nerea Aldunate, Cristian Brotfeld, Ernesto Guerra, Edmundo Kronmüller.

**Writing – original draft:** Nerea Aldunate, Pablo López-Silva, Edmundo Kronmüller.

**Writing – review & editing:** Nerea Aldunate, Pablo López-Silva, Ernesto Guerra, Edmundo Kronmüller.

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
