## [Decision Letter · Decision Letter 0]

12 Oct 2023

PONE-D-23-26275Spanish version of Multidimensional Mentalizing Questionnaire (MMQ): Translation, adaptation and psychometric properties in a Chilean populationPLOS ONE

Dear Dr. Kronmüller,

Thank you for submitting your manuscript to PLOS ONE. After careful consideration, we feel that it has merit but does not fully meet PLOS ONE’s publication criteria as it currently stands. Therefore, we invite you to submit a revised version of the manuscript that addresses the points raised during the review process.

While this is a very interesting topic, with promising results, several aspects of the writing still need to be improved. I urge you to address the corrections and indications suggested by the reviewers, which are aimed at obtaining a better article.

We look forward to receiving your revised manuscript.

Kind regards,

Juan-Luis Castillo-Navarrete, Ph.D.

Academic Editor

PLOS ONE

Journal Requirements:

This research was mainly funded by the Fondecyt-Chile Grant (1200655) from the Agencia Nacional de Investigación y Desarrollo de Chile (ANID) to EK. Additional support was provided by Fondecyt-Chile Grant (11190245) from the Agencia Nacional de In-vestigación y Desarrollo de Chile (ANID) to NA, and by Fondecyt-Chile Grant (1221058) from the Agencia Nacional de Investigación y Desarrollo de Chile (ANID) to PL. Funding from ANID/PIA/Basal Funds for Centers of Excellence Project FB0003 is also gratefully acknowledged. 

Reviewers' comments:

Reviewer's Responses to Questions

**Comments to the Author**

1. Is the manuscript technically sound, and do the data support the conclusions?

Reviewer #1: Partly

2. Has the statistical analysis been performed appropriately and rigorously? 

Reviewer #1: No

3. Have the authors made all data underlying the findings in their manuscript fully available?

Reviewer #1: Yes

4. Is the manuscript presented in an intelligible fashion and written in standard English?

Reviewer #1: Yes

5. Review Comments to the Author

Reviewer #1: Thank you for the invitation to review “Spanish version of Multidimensional Mentalizing Questionnaire (MMQ): Translation, adaptation and psychometric properties in a Chilean population.”

This paper reports on the translation and adaptation of the Multidimensional Mentalizing Questionnaire (MMQ) into Spanish for a native Spanish-speaking sample in Chile. The study examines the psychometric properties and internal consistency of the translated MMQ. Convergent evidence for validity is reported by correlating MMQ subscales with the Interpersonal Reactivity Index (IRI) and Empathy Quotient (EQ). Divergent evidence for validity is assessed by correlating aggressive traits using the Buss-Perry Aggression Questionnaire (BPAQ). The study also explores gender and age differences. Results reveal positive correlations between good mentalizing and empathy, particularly cognitive empathy, supporting the significance of positive mentalization in empathy. Negative mentalization is associated with difficulties in perspective-taking and social skills, as well as aggressive traits. Gender differences in mentalizing capacities are observed, and negative aspects of mentalization decrease with age.

The investigators are to be commended for undertaking such an ambitious project. However, I do have several suggestions for improving the ms., addressed below.

First, the authors write of “convergent validity,” and divergent validity.” There is only one validity, but there can be various sources of evidence for validity, such as convergent evidence and divergent evidence.

Second, the authors apparently conducted only one CFA, presumably testing a common factors model, whereas I recommend conducting an assessment of variance composition to determine if a total score can be modeled with alternative structures including bifactor, hierarchical, and unidimensional models. It appears that a common factors model was tested, in which responses to the MMQ were used as indicators of the hypothesized latent factors, with each item-level indicator loading on only one factor. The purpose of this model is to confirm the factor structure found in the EFA. Next, a bifactor model should be investigated. In the bifactor model, each item-level indicator is specified as having factor loadings on both a general factor (corresponding to the total scale score) and a group factor (corresponding to the subscale scores). All group factors should be specified as orthogonal to the general factor and to each other by fixing their intercorrelations to zero. A third alternative is the hierarchical model, in which the higher-order factor represents the general factor. The indicators of the higher-order factor are the lower-order latent variables corresponding to domain-specific constructs, in this case the subscales. The indicators for the lower-order latent constructs are the measured variables (e.g., item-level responses). Finally, a unidimensional model, in which all indicators load only on a general factor should be assessed.

Third, the authors compared different groups based on gender and age. Typically, with newly adapted scale one, to do this requires the assessment of whether or not the groups understand the scale in the same way. Whether or not particular scales are understood in the same way by men and women and younger and older people can be examined quantitatively through the assessment of measurement equivalence/invariance (ME/I) across age groups. This is a highly technical psychometric analysis that focusses on whether certain parameters resulting from confirmatory factor analyses are the same across the groups. When these parameters – factor loadings, item intercepts, and residuals – are the same for all groups, we can say that the different groups based on gender and age understand the scores of the scale in the same way, and we can reliably compare their men scores (Cheung and Lau, 2012; Kline, 2016). If they are not the same, then construct bias is likely present, and therefore we cannot have confidence in the results when comparing their scores on the scale. Assessing ME/I is clearly fundamental to good social science, but it is an oft-ignored prerequisite to comparing the scores of members of different groups on a scale, creating what a recent study has labeled “hidden invalidity” in the research literature (Hussey and Hughes, 2020).

Furthermore, because measurement invariance testing is limited by yielding only a binary decision (invariance or non-invariance), we recommend performing a series of post-hoc bootstrap confidence interval tests, as recommended by Cheung & Lau (2012), to examine the statistical significance of any between-group difference on parameters (factor loadings and item intercepts) if the Δχ2 or ΔCFI did not support invariance. We also recommend calculating the effect size for measurement non-invariance using the DMACS, the SEM equivalent of a Cohen’s d (effect size for means and covariance structures (MACS; Nye & Drasgow, 2011; Nye et al., 2018). Based on simulation data, in which researchers manipulated the level of non-invariance in a model from small (i.e., not practically meaningful) to large (Nye et al., 2018), DMACS were interpreted using the following effect size cutoffs: .40 (small), .60 (medium), and .80 and greater (large).

Finally, the authors investigate evidence for validity using raw scores. Since raw scores contain error; therefore, I recommend comparing latent scores using structural regression (in Structural Equation Modeling).

References

Cheung, G. W., Lau, R. S. 2012. “A Direct Comparison Approach for Testing Measurement Invariance.” Organizational Research Methods 15:167-198.

Hussy, I., Hughes, S. (2020). Hidden Invalidity Among 15 Commonly Used Measures in Social and Personality Psychology. Advances in Methods and Practices of Psychological Science 3:166-184.

Kline, R. B. 2016. Principles and Practice of Structural Equation Modeling (Fourth Ed.). New York: Guilford.

Nye, C. D., Bradburn, J., Olenick, J., Bialko, C., Drasgow, F. 2018. “How Big Are My Effects? Examining the Magnitude of Effect Sizes in Studies of Measurement Equivalence.” Organizational Research Methods 22: 678-709.

Nye, C. D., Drasgow, F. 2011. “Effect Size Indices for Analyses of Measurement Equivalence: Understanding the Practical Importance of Differences Between Groups.” Journal of Applied Psychology 96:966–980.

6. PLOS authors have the option to publish the peer review history of their article (what does this mean?). If published, this will include your full peer review and any attached files.

Reviewer #1: **Yes: **Ronald F. Levant

---

## [Author Response · Author response to Decision Letter 0]

5 Nov 2023

We thank our reviewer for his suggestions. We have addressed them all in the manuscript. With these changes and additions, our manuscript has substantially improved.

Reviewer #1: Thank you for the invitation to review “Spanish version of Multidimensional Mentalizing Questionnaire (MMQ): Translation, adaptation and psychometric properties in a Chilean population.”

This paper reports on the translation and adaptation of the Multidimensional Mentalizing Questionnaire (MMQ) into Spanish for a native Spanish-speaking sample in Chile. The study examines the psychometric properties and internal consistency of the translated MMQ. Convergent evidence for validity is reported by correlating MMQ subscales with the Interpersonal Reactivity Index (IRI) and Empathy Quotient (EQ). Divergent evidence for validity is assessed by correlating aggressive traits using the Buss-Perry Aggression Questionnaire (BPAQ). The study also explores gender and age differences. Results reveal positive correlations between good mentalizing and empathy, particularly cognitive empathy, supporting the significance of positive mentalization in empathy. Negative mentalization is associated with difficulties in perspective-taking and social skills, as well as aggressive traits. Gender differences in mentalizing capacities are observed, and negative aspects of mentalization decrease with age.

The investigators are to be commended for undertaking such an ambitious project. However, I do have several suggestions for improving the ms., addressed below.

First, the authors write of “convergent validity,” and divergent validity.” There is only one validity, but there can be various sources of evidence for validity, such as convergent evidence and divergent evidence.

Response 1:

We thank the reviewer’s comment. Following his suggestion, we modified the manuscript, referring to convergent and divergent evidence instead.

Second, the authors apparently conducted only one CFA, presumably testing a common factors model, whereas I recommend conducting an assessment of variance composition to determine if a total score can be modeled with alternative structures including bifactor, hierarchical, and unidimensional models. It appears that a common factors model was tested, in which responses to the MMQ were used as indicators of the hypothesized latent factors, with each item-level indicator loading on only one factor. The purpose of this model is to confirm the factor structure found in the EFA. Next, a bifactor model should be investigated. In the bifactor model, each item-level indicator is specified as having factor loadings on both a general factor (corresponding to the total scale score) and a group factor (corresponding to the subscale scores). All group factors should be specified as orthogonal to the general factor and to each other by fixing their intercorrelations to zero. A third alternative is the hierarchical model, in which the higher-order factor represents the general factor. The indicators of the higher-order factor are the lower-order latent variables corresponding to domain-specific constructs, in this case the subscales. The indicators for the lower-order latent constructs are the measured variables (e.g., item-level responses). Finally, a unidimensional model, in which all indicators load only on a general factor should be assessed.

Response 2:

Following the reviewer’s suggestion, to determine the best factor structure we tested 3 additional models and reported them in the manuscript: bifactor, one-factor, and hierarchical factor model. As can be seen in Table 3, our results show that common factors model had the best fit indexes compared to the 3 additional models.

Third, the authors compared different groups based on gender and age. Typically, with newly adapted scale one, to do this requires the assessment of whether or not the groups understand the scale in the same way. Whether or not particular scales are understood in the same way by men and women and younger and older people can be examined quantitatively through the assessment of measurement equivalence/invariance (ME/I) across age groups. This is a highly technical psychometric analysis that focusses on whether certain parameters resulting from confirmatory factor analyses are the same across the groups. When these parameters – factor loadings, item intercepts, and residuals – are the same for all groups, we can say that the different groups based on gender and age understand the scores of the scale in the same way, and we can reliably compare their men scores (Cheung and Lau, 2012; Kline, 2016). If they are not the same, then construct bias is likely present, and therefore we cannot have confidence in the results when comparing their scores on the scale. Assessing ME/I is clearly fundamental to good social science, but it is an oft-ignored prerequisite to comparing the scores of members of different groups on a scale, creating what a recent study has labeled “hidden invalidity” in the research literature (Hussey and Hughes, 2020).

Response 3:

A measurement invariance analysis was carried out using gender as group variable. We followed the recommendations of Wu and Estabrook (2016). These authors are concerned with how the threshold model is identified, and provide evidence to show that some constrains are needed to test the hypothesis that the instrument is measuring the same way different groups.

Ref: Wu, H. & Estabrook, R. (2016). Identification of Confirmatory Factor Analysis Models of Different Levels of Invariance for Ordered Categorical Outcomes. Psychometrika, 81(4), 1014–1045. doi:10.1007/s11336-016-9506-0

With respect to the age-based invariance analysis, our sample is predominantly young, so there is no clear balance with respect to the possible groups (see Figure below). If, for example, the median is used, one group would be homogeneous in terms of age, while another (the older group) would not. In addition, since we are not dealing with natural groups in our sample, we have no clear criteria to establish a cut-off point by age. We do acknowledge, however, that an analysis of invariance is important to draw conclusions regarding the correlation between the MMQ and age.

Furthermore, because measurement invariance testing is limited by yielding only a binary decision (invariance or non-invariance), we recommend performing a series of post-hoc bootstrap confidence interval tests, as recommended by Cheung & Lau (2012), to examine the statistical significance of any between-group difference on parameters (factor loadings and item intercepts) if the Δχ2 or ΔCFI did not support invariance. We also recommend calculating the effect size for measurement non-invariance using the DMACS, the SEM equivalent of a Cohen’s d (effect size for means and covariance structures (MACS; Nye & Drasgow, 2011; Nye et al., 2018). Based on simulation data, in which researchers manipulated the level of non-invariance in a model from small (i.e., not practically meaningful) to large (Nye et al., 2018), DMACS were interpreted using the following effect size cutoffs: .40 (small), .60 (medium), and .80 and greater (large).

Response 4:

Considering that our analysis of invariance showed no differences across gender, we did not perform these analyses.

Finally, the authors investigate evidence for validity using raw scores. Since raw scores contain error; therefore, I recommend comparing latent scores using structural regression (in Structural Equation Modeling).

Response 5:

In the revised manuscript, we report the correlations between the subscales of the MMQ with the latent scores and the raw scores. There were some changes in magnitude but not in direction. In the case of the correlations between the other instruments and the subscales of the MMQ, we report both the correlation between the latent scores of the MMQ and the raw scores of the other scales and between the raw scores of all scales and subscales. 

References

Cheung, G. W., Lau, R. S. 2012. “A Direct Comparison Approach for Testing Measurement Invariance.” Organizational Research Methods 15:167-198.

Hussy, I., Hughes, S. (2020). Hidden Invalidity Among 15 Commonly Used Measures in Social and Personality Psychology. Advances in Methods and Practices of Psychological Science 3:166-184.

Kline, R. B. 2016. Principles and Practice of Structural Equation Modeling (Fourth Ed.). New York: Guilford.

Nye, C. D., Bradburn, J., Olenick, J., Bialko, C., Drasgow, F. 2018. “How Big Are My Effects? Examining the Magnitude of Effect Sizes in Studies of Measurement Equivalence.” Organizational Research Methods 22: 678-709.

Nye, C. D., Drasgow, F. 2011. “Effect Size Indices for Analyses of Measurement Equivalence: Understanding the Practical Importance of Differences Between Groups.” Journal of Applied Psychology 96:966–980.

---

## [Decision Letter · Decision Letter 1]

23 Nov 2023

PONE-D-23-26275R1Spanish version of Multidimensional Mentalizing Questionnaire (MMQ): Translation, adaptation and psychometric properties in a Chilean populationPLOS ONE

Dear Dr. Kronmüller,

Thank you for submitting your manuscript to PLOS ONE. After careful consideration, we feel that it has merit but does not fully meet PLOS ONE’s publication criteria as it currently stands. Therefore, we invite you to submit a revised version of the manuscript that addresses the points raised during the review process.

**ACADEMIC EDITOR: Please insert comments here and delete this placeholder text when finished.** Be sure to:

I would like to highlight the work carried out, which has led to a substantial improvement in the writing. This study represents an important advance in the adaptation and validation of psychometric instruments in specific cultural contexts. 

In light of the above, I would like to outline some minor suggestions aimed at enriching the paper.

Summary and State of the Art

The summary provides a clear and concise overview of the purpose of the study, its methodology, and the main findings. The contextualisation of the MMQ and the rationale for its adaptation to Chilean Spanish are well supported. However, a more detailed discussion of how the MMQ compares and contrasts with similar instruments already existing in the Spanish-speaking context is missing, which would strengthen the rationale for this specific adaptation.

Problematisation and Research Question

The problematisation and formulation of the research question are adequate. The need for a validated instrument to measure mentalisation in the Chilean population is clearly identified. However, it would be beneficial to further explore how this study specifically contributes to clinical practice and mental health research in Chile.

Hypotheses and Objectives

The objectives are clear and relevant, focusing on translating, adapting and evaluating the psychometric properties of the MMQ. However, a more specific hypothesis about the expectations of adaptation, especially in terms of cultural constructs or demographic differences, could have added depth to the study.

Methodology

The methodology is rigorous, with a detailed focus on translation, adaptation and analysis of the factor structure of the MMQ. The inclusion of measurement invariance analysis to examine gender differences is a strength. However, the predominantly young sample and the absence of an age-based invariance analysis limit the generalisability of the results. In addition, it would be useful to elaborate more on how item deletion decisions were handled and justified during adaptation. This is in order to safeguard the merodological and statistical process.

Discussion and Conclusions

The discussion is comprehensive in terms of the interpretation of the results, highlighting the relationships between mentalisation, empathy and aggressive traits. Consideration of limitations and suggestions for future research is adequate. However, it would be valuable to include a more detailed discussion of the practical implications of these findings for mental health professionals in Chile.

Concluding comment

The article is valuable and contributes significantly to the psychometric literature, especially in the context of mental health in Chile. The corrections made by the authors have substantially improved the manuscript, so these minor corrections, particularly with regard to the discussion of the specific cultural context and the generalisation of the results, will allow for an adequate writing.

We look forward to receiving your revised manuscript.

Kind regards,

Juan Luis Castillo-Navarrete, Ph.D.

Academic Editor

PLOS ONE

 Journal Requirements:

Reviewers' comments:

Reviewer's Responses to Questions

**Comments to the Author**

1. If the authors have adequately addressed your comments raised in a previous round of review and you feel that this manuscript is now acceptable for publication, you may indicate that here to bypass the “Comments to the Author” section, enter your conflict of interest statement in the “Confidential to Editor” section, and submit your "Accept" recommendation.

Reviewer #1: All comments have been addressed

2. Is the manuscript technically sound, and do the data support the conclusions?

Reviewer #1: Yes

3. Has the statistical analysis been performed appropriately and rigorously? 

Reviewer #1: Yes

4. Have the authors made all data underlying the findings in their manuscript fully available?

Reviewer #1: Yes

5. Is the manuscript presented in an intelligible fashion and written in standard English?

Reviewer #1: Yes

6. Review Comments to the Author

Reviewer #1: Thank you for the invitation to review the first revision (R1) of the “Spanish version of Multidimensional Mentalizing Questionnaire (MMQ): Translation, adaptation and psychometric properties in a Chilean population.”

This paper reports on the translation and adaptation of the Multidimensional Mentalizing Questionnaire (MMQ) into Spanish for a native Spanish-speaking sample in Chile. The study examines the psychometric properties and internal consistency of the translated MMQ. Convergent evidence for validity is reported by correlating MMQ subscales with the Interpersonal Reactivity Index (IRI) and Empathy Quotient (EQ). Divergent evidence for validity is assessed by correlating aggressive traits using the Buss-Perry Aggression Questionnaire (BPAQ). The study also explores gender and age differences. Results reveal positive correlations between good mentalizing and empathy, particularly cognitive empathy, supporting the significance of positive mentalization in empathy. Negative mentalization is associated with difficulties in perspective-taking and social skills, as well as aggressive traits. Gender differences in mentalizing capacities are observed, and negative aspects of mentalization decrease with age.

The authors have responded to all of my suggestions.

7. PLOS authors have the option to publish the peer review history of their article (what does this mean?). If published, this will include your full peer review and any attached files.

Reviewer #1: **Yes: **Ronald F. Levant

---

## [Author Response · Author response to Decision Letter 1]

14 Dec 2023

We thank the editor for his comments and suggestions, which will surely contribute to the quality of our work. We have addressed each one in the text (in red font). We specify those additions in what follows, within quotes and in italics, linking them to each comment.

1.- Summary and State of the Art

The summary provides a clear and concise overview of the purpose of the study, its methodology, and the main findings. The contextualisation of the MMQ and the rationale for its adaptation to Chilean Spanish are well supported. However, a more detailed discussion of how the MMQ compares and contrasts with similar instruments already existing in the Spanish-speaking context is missing, which would strengthen the rationale for this specific adaptation.

ANSWER: We thank the editor for the comment. We have included in the abstract a reference to the fact that this would be the first scale adapted to the Chilean population that measures, through self-reporting, people's perception in their daily experiences with mentalization. We have further elaborated on this in the article, specifically following the editor’s comments 2, 3, and 5.

“The availability of the Spanish translation of the MMQ, the first self-reporting scale measuring mentalization adapted to Chilean population, contributes to research aiming to understand its relationship with other psychological phenomena in different cultural context and facilitating clinical interventions with different population groups. We therefore encourage further investigation into cultural, gender and age differences in mentalization.”

2.- Problematisation and Research Question

The problematisation and formulation of the research question are adequate. The need for a validated instrument to measure mentalisation in the Chilean population is clearly identified. However, it would be beneficial to further explore how this study specifically contributes to clinical practice and mental health research in Chile.

ANSWER: In the introduction, we have elaborated on the contributions of the MMQ in the clinical and mental health context.

“Indeed, as mentioned above, mentalization is associated with mental health at the level of relational difficulties and interpersonal stress in schizophrenia and antisocial and borderline personality disorders. Moreover, suicidal behavior could also be linked to a low capacity for mentalization (44, 45). Thus, early detection of low levels of mentalization could serve both prevention and management in therapeutic interventions such as Mentalization-Based Therapy (46-48). In this regard, it has also been proposed that mentalization is one of the primary tools for establishing therapeutic settings that could lead to a good alliance between therapist and patient. Finally, the MMQ could also be applied to therapists in training, serving to identify shortcomings and recommend the development of mentalizing skills (47, 49, 50).”

3.- Hypotheses and Objectives

The objectives are clear and relevant, focusing on translating, adapting and evaluating the psychometric properties of the MMQ. However, a more specific hypothesis about the expectations of adaptation, especially in terms of cultural constructs or demographic differences, could have added depth to the study.

ANSWER: As the study's focus was exclusively on examining the psychometric properties of the Chilean adaptation of the MMQ, we did not formulate specific objectives or hypotheses for cross-cultural comparisons. Additionally, exploratory analyses were conducted for gender and age comparisons. One limitation of our study is the age distribution within the sample, which precludes conclusive results due to a significant decrease in sample size as age progresses. We address these limitations in greater detail in the discussion, emphasizing their relevance to future research considerations.

“In conclusion, the Spanish adaptation of the MMQ presented in this work demonstrates robust psychometric properties in the Chilean population. The study provides evidence of how the different dimensions of mentalizing may be related to empathic abilities and aggressive behaviors. Considering that the original version was developed in Italian, it is crucial to address potential cultural differences, as they may influence the behavioral expression of mentalization and item functioning (85, 86). Therefore, we advocate for cross-cultural studies that, among other questions, test cultural invariance. Additionally, exploring the relationship between age and mentalization and other pertinent demographic analyses in forthcoming investigations is crucial. Finally, the adaptation of the MMQ presented in this paper is the first tool that will serve for self-administered measurements of mentalization in Chile, providing a standardized and valid measure that can inform psychological treatment and research.”

4.- Methodology

The methodology is rigorous, with a detailed focus on translation, adaptation and analysis of the factor structure of the MMQ. The inclusion of measurement invariance analysis to examine gender differences is a strength. However, the predominantly young sample and the absence of an age-based invariance analysis limit the generalisability of the results. In addition, it would be useful to elaborate more on how item deletion decisions were handled and justified during adaptation. This is in order to safeguard the methodological and statistical process.

ANSWER: We thank the editor for these observations. To address the first one, we added a caveat in the discussion as follows:

“Moreover, future work should address the generalizability of our findings across life span through invariance analysis.” 

ANSWER: Also, we added the following text to address the second comment, further explaining the rationale for modifying the original scale by not including five items.

“Aiming to improve the fit and, at the same time, safeguard the simplicity and the parsimony of the model, we look for items that correlated with one or more additional factors different than the one to which they were initially associated based on theoretical and empirical grounds. Specifically, we examined the modification indices provided by the Lavaan package v. 0.6.15 (61) to identify items to which adding an extra parameter, i.e., a factor weight, substantially improved the fit indicators of the model. Following this rationale, five items exhibited substantial cross-loadings (18, 26, 30, 31, and 32). Three of these items originally belonged to the Reflexivity subscale, which had the largest number of items, and the remaining two belonged to Ego Strength, the second largest subscale. As shown in Table 3, the model’s fit indicators improved substantially by removing those items, placing them within the accepted psychometric standards for the common factors model (62).”

5.- Discussion and Conclusions

The discussion is comprehensive in terms of the interpretation of the results, highlighting the relationships between mentalisation, empathy and aggressive traits. Consideration of limitations and suggestions for future research is adequate. However, it would be valuable to include a more detailed discussion of the practical implications of these findings for mental health professionals in Chile.

ANSWER: In the discussion, we have elaborated the practical implications of the MMQ results in the clinical and mental health context.

“In conclusion, the Spanish adaptation of the MMQ presented in this work demonstrates robust psychometric properties in the Chilean population. The study provides evidence of how the different dimensions of mentalizing may be related to empathic abilities and aggressive behaviors. Considering that the original version was developed in Italian, it is crucial to address potential cultural differences, as they may influence the behavioral expression of mentalization and item functioning (85, 86). Therefore, we advocate for cross-cultural studies that, among other questions, test cultural invariance. Additionally, exploring the relationship between age and mentalization and other pertinent demographic analyses in forthcoming investigations is crucial. Finally, the adaptation of the MMQ presented in this paper is the first tool that will serve for self-administered measurements of mentalization in Chile, providing a standardized and valid measure that can inform psychological treatment and research.”

6.- Concluding comment

The article is valuable and contributes significantly to the psychometric literature, especially in the context of mental health in Chile. The corrections made by the authors have substantially improved the manuscript, so these minor corrections, particularly with regard to the discussion of the specific cultural context and the generalisation of the results, will allow for an adequate writing.

We thank the editor for this comment and totally agree with it. We hope we have addressed all issues raised.

---

## [Editor Report · Decision Letter 2]

18 Dec 2023

Spanish version of Multidimensional Mentalizing Questionnaire (MMQ): Translation, adaptation and psychometric properties in a Chilean population

PONE-D-23-26275R2

Dear Dr. Edmundo Kronmüller,

We’re pleased to inform you that your manuscript has been judged scientifically suitable for publication and will be formally accepted for publication once it meets all outstanding technical requirements.

Kind regards,

Juan Luis Castillo-Navarrete, Ph.D.

Academic Editor

PLOS ONE

Additional Editor Comments (optional):

My congratulations to the authors for the improvements made to the paper, which represents a considerable contribution to the study of the subject and especially to the development of mental health, not only in their country, but also globally.
---

## [Editor Report · Acceptance letter]

28 Dec 2023

PONE-D-23-26275R2 

PLOS ONE

Dear Dr. Kronmüller, 

I'm pleased to inform you that your manuscript has been deemed suitable for publication in PLOS ONE. Congratulations! Your manuscript is now being handed over to our production team.

Kind regards, 

on behalf of

Dr. Juan Luis Castillo-Navarrete 

Academic Editor

PLOS ONE